# Collapse of Intra-Tumor Cooperation Induced by Engineered Defector Cells

**DOI:** 10.3390/cancers13153674

**Published:** 2021-07-22

**Authors:** Marco Archetti

**Affiliations:** Department of Biology, Pennsylvania State University, University Park, State College, PA 16802, USA; mua972@psu.edu

**Keywords:** cooperation, competition, resistance, treatment, therapy, evolution, game theory, growth factor, dynamics

## Abstract

**Simple Summary:**

Using experiments in vitro and a mathematical model, I show that genetically modified cancer cells in which the gene for an essential growth factor is knocked out can spread in a population of cells that produce that growth factor, leading to an overall reduction in proliferation—a proof of principle for a potential treatment that harnesses clonal selection by impairing intra-tumor cooperation.

**Abstract:**

Anti-cancer therapies promote clonal selection of resistant cells that evade treatment. Effective therapy must be stable against the evolution of resistance. A potential strategy based on concepts from evolutionary game theory is to impair intra-tumor cooperation using genetically modified cells in which genes coding for essential growth factors have been knocked out. Such engineered cells would spread by clonal selection, driving the collapse of intra-tumor cooperation and a consequent reduction in tumor growth. Here, I test this idea in vitro in four cancer types (neuroendocrine pancreatic cancer, mesothelioma, lung adenocarcinoma and multiple myeloma). A reduction, or even complete eradication, of the producer clone and the consequent reduction in cell proliferation, is achieved in some but not all cases by introducing a small fraction of non-producer cells in the population. I show that the collapse of intra-tumor cooperation depends on the cost/benefit ratio of growth factor production. When stable cooperation among producer and non-producer cells occurs, its collapse can be induced by increasing the number of growth factors available to the cells. Considerations on nonlinear dynamics in the framework of evolutionary game theory explain this as the result of perturbation of the equilibrium of a system that resembles a public goods game, in which the production of growth factors is a cooperative phenotype. Inducing collapse of intra-tumor cooperation by engineering cancer cells will require the identification of growth factors that are essential for the tumor and that have a high cost of production for the cell.

## 1. Introduction

Cancer treatment failure is often caused by the evolution of resistant clones within a tumor due to the clonal selection of mutant cells that evade therapy. These clones can proliferate faster than the original susceptible ones, spreading within the tumor and eventually leading to population-wide resistance. While this process is understood to be analogous to Darwinian selection [1,2,3,4], applications of concepts and methods from evolutionary biology are still rare in cancer research [5]. In fact, administering drugs at the maximum tolerated dose—the conventional approach adopted in cancer therapy—has no compelling justification from the point of view of evolutionary dynamics [6,7,8] because it eliminates potential competitors and can lead to the rapid proliferation of resistant clones. Recent less conventional treatment strategies have applied evolutionary principles to inhibit the emergence of treatment-resistant populations by the appropriately timed withdrawal of treatment [9,10,11]. The logic of this “adaptive therapy” is to harness clonal selection—the very force responsible for the evolution of resistance to conventional therapies—by enabling competition between clones. 

Another way to harnesses clonal selection for therapy is to disrupt intra-tumor cooperation. Cancer cells not only compete for space and resources within the tumor but also cooperate by producing and sharing diffusible molecules such as growth factors that promote proliferation, enable evasion from apoptosis and the immune system and induce neo-angiogenesis. Since growth factors diffuse in the extracellular matrix, their effects are not limited to producer cells, and growth factor production can be considered a form of cooperation between cancer cells [12,13,14,15,16,17,18,19,20,21,22,23,24]. Mutant cancer cells that do not produce the growth factor, or that produce it in lower amounts, can still benefit from the growth factors secreted by their neighboring cells, and under certain circumstances, both types can coexist by negative frequency-dependent clonal selection [20,21,22,23,24,25,26]. Evolutionary game theory suggests that if the cost of producing the growth factor is high enough, however, non-producer cells can have a proliferation advantage (due to the lack of cost for producing the growth factor) that enables them to increase in frequency within the tumor, eventually leading to the extinction of the producer clone and the collapse of cooperation [20,21,22,23,24,25]. If autologous cancer cells could be engineered by deleting genes for essential growth factors, these genetically modified cells could spread within the original tumor by clonal selection, leading to the collapse of intra-tumor cooperation and a reduction in the production of the growth factors [22,23,24,25]. 

Here, I provide a proof of principle of this autologous cell defection approach in four cell lines in which a growth factor was knocked out: IGF2 (insulin-like growth factor 2) in βTC neuroendocrine pancreatic cancer cells (βTC-ΔIGF2 clone); IL6 (interleukin 6) in U266B1 multiple myeloma cells (U266B1-ΔIL6 clone) and in A549 lung cancer cells (A549-ΔIL6 clone); and PDGFD (platelet-derived growth factor D) in NCIH28 mesothelioma cells (NCIH28-ΔPDGFD clone). These combinations of cell lines and growth factors were chosen based on existing evidence (reviewed in the Discussion section) that the growth factor affects proliferation. I show, combining the results of experiments in vitro with knockout (KO) and wild-type (WT) clones and a mathematical model in the framework of evolutionary game theory, that the cost of growth factor production and the benefit it confers to the tumor are critical in driving the collapse of cooperation, and I discuss the potential implications for therapy.

## 2. Results

### 2.1. Long-Term Evolution of Mixed Populations: KO Cells Can Induce the Collapse of Cooperation or a Stable Polymorphism 

A small fraction (1%) of KO cells were introduced in WT populations (200,000 cells) and passaged (split when confluent) 12 times, tracking changes in the fraction of cell types (Figure 1). Three different amounts of FBS (fetal bovine serum) in the growth medium were used to modify the relative benefit of endogenous (WT-produced) growth factor (as FBS increases, the benefit of producing endogenous growth factor decreases). 

The βTC-ΔIGF2 clone invaded βTC beta cells populations when growing with 10% FBS, virtually driving the WT clone to extinction. At 5% FBS, instead, the two clones seemed to reach a stable equilibrium, and at 2% FBS, the KO clone was unable to invade the WT population.The U266B1-ΔIL6 clone was always able to invade the U266B1 population, replacing the WT clone at 5% and 10% FBS and reaching a stable mixed equilibrium at 2% FBS.The A549-ΔIL6 clone was also always able to invade the WT A549 clone, reaching a stable mixed equilibrium for any amount of FBS.The NCIH28-ΔPDGFD clone was never able to invade the WT population at low amounts of FBS but showed a slight change from 1% to 9% (*t*_3_ = 11.3, *p* = 0.0015) at 10% FBS, perhaps a sign of a slow increase.

### 2.2. Growth Rates of Final Mixed Populations: Collapse of Cooperation Impairs Proliferation

The growth rates of the mixed populations were also measured at the first and last passage (Figure 1) to test whether increases in the fraction of the KO cells lead to an overall decline of proliferation. The growth rates of the final populations were lower than the growth rates of the initial populations in βTC and U266B1 cells (at 10% FBS and 5% FBS, but not significantly at 2% FBS), but not in NCIH28 and A549 cells. 

βTC: at 10% FBS, growth was 72% slower (*t*_5.9_ = 14.4, *p* = 8 × 10^−6^); at 5% FBS, it was 65% slower (*t*_10_ = 13.0, *p* = 1 × 10^−7^); at 2% FBS, it was 25% slower (*t*_10_ = 1.2, *p* = 0.25).U266B1: at 10% FBS, growth was 50% slower (*t*_10_ = 7.2, *p* = 3 × 10^−5^); at 5% FBS, it was 57% slower (*t*_10_ = 7.6, *p* = 2 × 10^−5^); at 2% FBS, it was 18% slower (*t*_10_ = 1.2, *p* = 0.27).A549: differences in growth rates between initial and final populations were not significant at 10% FBS (*t*_10_ = 1.0, *p* = 0.35), 5% FBS (*t*_10_ = 1.7, *p* = 0.12) or 2% FBS (*t*_10_ = 2.1, *p* = 0.07).NCIH28: differences in growth rates between initial and final populations were not significant at 10% FBS (*t*_10_ = 0.3, *p* = 0.75), 5% FBS (*t*_10_ = 0.3, *p* = 0.73) or 2% FBS (*t*_10_ = 0.7, *p* = 0.50).

Therefore, extinction of the WT clone, when it occurred, always led to a significant reduction in growth rates. The mixed equilibrium observed in βTC cells at 5% FBS was the only case in which the coexistence of the two types was accompanied by a significant reduction in cell growth; in the other cell lines, the coexistence of the two types did not lead to a significant reduction in growth rates. 

### 2.3. Growth Rates of Pure Populations: Estimating the Cost and Benefit of Cooperation

In order to understand the differences in the dynamics observed in the different cell lines, the growth rates of pure populations were measured (with 5% FBS). Using increasing amounts of soluble recombinant growth factors, it is possible to estimate the costs and benefits of the growth factors. Pure KOs are expected to grow slower than WT populations in the absence of the growth factor. In the presence of a saturating amount of growth factor, the KO clone may grow faster than the WT clone (because the lack of endogenous production of growth factor is compensated by the recombinant growth factor in the growth medium, and because the KO clone does not pay the cost of producing it). 

In all cases, the KO clone grew significantly less than the WT clone in the absence of exogenous growth factor (Figure 2):βTC: 79% less (*t*_10_ = 13.0, *p* = 1 × 10^−7^);U266B1: 55% less (*t*_10_ = 7.1, *p* = 3 × 10^−5^);A549: 39% less (*t*_10_ = 8.3, *p* = 8 × 10^−6^);NCIH28: 79% less (*t*_10_ = 11.3, *p* = 5 × 10^−7^).

βTC and NCIH28 have a relatively high dependency on the growth factor that was deleted (IGF2 and PDGFD, respectively), as they can grow only at 21% the rate of the WT clones in the absence of recombinant growth factors. U266B1 and especially A549 are less dependent on the growth factor (IL6 in both cases) that was deleted, as the KO clones grow at, respectively, 45% and 61% the rate of the corresponding WT clones in the absence of exogenous growth factor. The baseline fitness W_0_ is therefore 0.21 for βTC and NCIH28, 0.45 for U266B1, and 0.61 for A549.

The KO clone, however, grew faster than the WT clone in the presence of saturating amounts of recombinant growth factor in the medium (but significantly faster only in βTC and in U266B1) (Figure 2):βTC: 20% more (*t*_10_ = 5.1, *p* = 0.00047);U266B1: 26% more (*t*_10_ = 4.9, *p* = 0.00065);A549: 2% more (*t*_10_ = 0.4, *p* = 0.71);NCIH28: 2% more (*t*_10_ = 1.3, *p* = 0.89).

Recombinant growth factor added to the growth medium generally does not significantly affect the growth of the WT clones (that already produce the growth factor) but improves the growth of the KO clones. Indeed, the KO clone grows faster than the WT clone for βTC and U266B1 (while the differences between WT and KO are not significant in A549 and NCIH28). U266B1-ΔIL6 cells seem to grow even in the absence of the growth factor, whereas knocking out IGF2 in βTC seems to have a more significant effect on proliferation. On the other hand, in NCIH28 and A549, saturating amounts of the growth factor improve the growth of the KO clone without conferring it, a significant advantage over the WT clone (Figure 2). These results point to a high cost of growth factor production for βTC and U266B1, but not for NCIH28 and A549, and to a high benefit conferred by the growth factors to βTC-ΔIGF2 and NCIH28-ΔPDGFD, and to a lesser extent U266B1-ΔIL6, but not A549-ΔIL6, which can grow efficiently even in the absence of exogenous growth factor. 

The cost and benefit of growth factor production can be estimated from the growth of pure populations (WT or KO) seeded at equal densities after the same time. The cost is estimated by *c* = S_KO_ − S_WT_, the difference in growth between WT, producer cells (S_WT_) and KO, non-producer cells (S_KO_) in the presence of saturating amounts of exogenous growth factor (because in that case, there is no benefit in producing extra endogenous growth factor—hence the only difference in growth must be due to the cost of secreting the cooperative molecules). The benefit of endogenously produced (by WT cells) growth factor is estimated by the difference in growth, in the absence of exogenous growth factor, between WT, producer cells (N_WT_) and KO, non-producer cells (N_KO_) because in that case, the only difference in growth must be due to the benefit of the growth factor. This difference in growth (N_WT_-N_KO_) must be equal to the benefit of the growth factor *minus the cost* of production; hence, the benefit is *b* = (N_WT_ − N_KO_) + *c* = (N_WT_ − N_KO_) + (S_KO_ − S_WT_). (If—but this is not necessarily true—the WT cells produced a saturating amount of the growth factor, then N_WT_ = S_WT_, and more simply, we would have *b* = S_KO_ − N_KO_, the difference in growth between KO cells at saturating levels of the growth factor and in its absence.) Estimating costs and benefits in this way, we can derive the cost/benefit ratio of growth factor production *c*/*b* = (S_KO_ − S_WT_)/(N_WT_ − N_KO_ + S_KO_ − S_WT_). The relative cost/benefit of growth factor production is high in U266B1 and βTC, low in A549 and NCIH28:βTC: *c/b =* 0.28U266B1: *c/b =* 0.33A549: *c/b =* 0.04NCIH28: *c/b =* 0.02

These results may explain the difference in the dynamics observed in the long-term evolution experiments (Figure 1) (at 5% FBS). 

*High benefit and low cost* (NCIH28). Even though a population of KO cells would have a significant reduction in proliferation, the low cost of growth factor production results in weak selection, hence no change in frequency: the KO clone simply does not spread (or perhaps not fast enough to be visible during the long-term experiment reported here) to induce a (potentially significant) reduction in proliferation.*Low benefit and low cost* (A549). In this case, even though the cost of growth factor production is low, the benefit is also modest (a population of KO cells would still be able to grow relatively well), and the resulting *c*/*b* ratio is high enough to drive the spread of the KO clone. The result is a stable coexistence of the KO and WT cells.*Low benefit and high cost* (U266B1). A high *c*/*b* ratio results in an efficient drive of the KO clone into the WT population and in the ultimate extinction of the WT clone. The lack of IL6, however, is not enough to induce a substantial impairment of proliferation.*High benefit and high cost* (βTC). High costs and benefits result, in this case, in a high *c*/*b* ratio (hence, an efficient drive of the KO clone into the WT population) which, even when (at 5% FBS) the WT and KO cells coexist, results in a significant reduction in average proliferation.

### 2.4. Additional Growth Factors Induce a Stable Collapse of Cooperation 

If, as we have seen so far, a few KO cells introduced in a population of WT cells spread to reach a mixed equilibrium (a mixture of KO and WT cells) at low values of FBS, and to reach fixation (drive the WT clone to extinction) at higher FBS levels, it stands to reason that a population at a mixed equilibrium could also be tipped to drive the WT clone to extinction by increasing the amount of FBS in the growth medium. This indeed seems to be the case (Figure 3).

Counterintuitively, better growing conditions (more FBS) lead to lower growth rates—a result that can be understood taking into account the reduction in the fraction of WT cells. A shift from a mixed equilibrium to the extinction of the WT clone induced by higher amounts of FBS is consistent with the result (Figure 2) that a high amount of FBS induces collapse of cooperation (extinction of the WT type) while a lower amount leads to the coexistence of the two types. What is apparently surprising here is that the shift is irreversible even if the WT has not gone extinct yet: reducing that amount of FBS back to the initial (lower) value does not bring the population back to its original mixed equilibrium; instead, the decline in the fraction of WT cells continues, leading to the virtual extinction of the WT clone (Figure 3) under conditions (amount of FBS) that would lead to a stable coexistence of the two types, had the process started from only a few KO cells (Figure 3). In other words, a transient perturbation, if strong enough, of a stable mixed population can induce a self-promoting collapse of cooperation that leads to the extinction of the WT clone.

### 2.5. Evolutionary Game Theory Reveals the Importance of Costs and Benefits, and Why the Collapse of Cooperation Is Self-Promoting

The fact that high amounts of FBS in the growth medium can lead to the collapse of cooperation, whereas lower amounts can lead to a stable coexistence of the two types (Figure 1) and can be ascribed to changes in the cost/benefit of growth factor production. In this section, I prove this point—the existence of a tipping point for *c*/*b* that leads from stable cooperation to its collapse—using a mathematical model in the framework of evolutionary game theory. This will also help explain the apparently surprising result that a transient perturbation of the stable mixed population can induce a self-promoting collapse of cooperation and the extinction of the WT clone (Figure 3). 

The coexistence of the two clones at lower amounts of FBS can be explained by the nonlinear effect of the growth factor on cell fitness (Figure 4A), which at low cost/benefit ratios produce a mixed equilibrium in which the two types coexist (Figure 4B) while at high cost/benefit ratio leads to the extinction of the WT clone (Figure 4C). Note that the details depend on the shape of the benefit function, hence on the effect of the growth factor on cell fitness: if the inflection point of the benefit function is high enough (a higher amount of growth factor is necessary to achieve the same benefit level (Figure 4D)), there can be no stable mixed equilibrium, and the population can remain at the pure cooperative equilibrium (the KO clone is unable to invade) even if producing the growth factor has a cost (Figure 4E). If the cost/benefit ratio is high enough, however, cooperation will collapse even in this case (Figure 4F). 

The model can also explain the apparently puzzling result that when the fraction of KO cells increase above a critical level, as a result of a transient increase of FBS, when the original amount of FBS is restored, cooperation collapses and the population does not bounce back to the original mixed equilibrium (Figure 3). If the effect of the growth factor is a typical sigmoid function (Figure 4A), when the fraction of WT cells is below a critical threshold, in the basin of attraction of the stable pure KO equilibrium, it cannot converge again to the original stable mixed equilibrium (that was reachable starting from a small fraction of KO cells). This tipping point (the unstable equilibrium x_U_ in Figure 4B,E) depends, again, on the shape of the benefit function. If the inflection point of the benefit function is low (that is, only a few cells are enough to produce a large benefit (Figure 4G)), it is possible that there is no unstable equilibrium (Figure 4H); in this case, cooperation will only collapse if the cost/benefit ratio is above the critical level (Figure 4I).

Therefore, the diversity of outcomes in the long-term dynamics (Section 3.1; Figure 1) can be explained by the difference in costs and benefits that we have described (Section 3.2; Figure 2) as explained in Figure 4. For a given shape of the benefit function, if the cost/benefit ratio of being a producer is higher than a critical threshold c_MAX_, WT producer cells are driven to extinction by KO non-producer cells. Below that critical threshold, WT and KO cells will coexist in a stable mixed equilibrium, irrespective of their initial frequencies, as an effect of negative frequency-dependent selection. If, however, the fraction of WT cells is below a critical level x_MIN_, the KO cells will drive the WT cells to extinction even when the cost is low. There are, therefore, two instances in which cooperation can collapse: the cost/benefit ratio is enough, or the fraction of KO cells is high enough.

## 3. Discussion

### 3.1. Summary of the Results

In summary:Non-producer cells can spread in a population of producer cells and induce the collapse of cooperation;The collapse of cooperation requires a high cost/benefit ratio of growth factor production;The collapse of cooperation impairs proliferation if the benefit conferred by the growth factor is high enough;Improving the growing conditions of the cell culture can induce the collapse of cooperation;The collapse of cooperation, if it occurs, is stable and self-promoting.

### 3.2. The Cost/Benefit Ratio of the Growth Factor Determines the Dynamics

The diversity of outcomes observed in the long-term experiments (Figure 1) requires further discussion. The first issue is that in some cases, the WT clone does not go extinct, remaining instead at a stable mixed equilibrium with the KO clone. This is predicted by evolutionary game theory for systems in which the benefit of the growth factor is a nonlinear function of its concentration [20,21,22,23,24,25,26], which is likely to be the case for all growth factors (and all biological molecules [27,28,29,30,31,32]). As we have seen, indeed, the cost/benefit ratio of the growth factor is a decisive determinant of the dynamics: a low cost/benefit leads to a stable coexistence of the two types, and the collapse of cooperation leads to significant lower proliferation only if the benefit is high enough. 

Incidentally, this result suggests that cell defection may be happening continuously in tumors, where spontaneous defector clones could spread by clonal selection (these mutant clones need not be complete KO: the dynamics would be similar for mutant clones that produce the growth factor in lower amounts, and depend on how large the cost/benefit ratio is). The few tumors that manage to grow (the probability of success for a growing tumor is very small [33,34]) are arguably the ones that produce growth factors with a low cost/benefit ratio. 

The implication of this is that genetically modified cells would be able to drive the collapse of cooperation, hence have potential therapeutic value, only if this cost/benefit ratio is high enough and the benefit is high enough. While in our experiments the benefit (hence the relative cost) could be modified by adding exogenous FBS or growth factor, this seems an undesirable complication for treatment—being able to drive intra-tumor cooperation to collapse simply by knocking out a few cells within the tumor would be preferable. Yet, the fact that adding growth factors to a population of cancer cells *reduces* its proliferation, which seems counterintuitive, is an interesting result that resembles the logic of adaptive therapy [9,10,11]: allowing the tumor to grow efficiently, in our case by providing additional growth factors it relies on (in the case of adaptive therapy by using a lower dosage of a drug) may ultimately be a better therapeutic approach. In our case, this is due to the increase in the frequency of KO cells induced by the increase in available growth factors, which is equivalent to increasing the cost/benefit ratio (in the case of adaptive therapy, the effect is due, instead, to the stronger competition among cells enabled by lowed dosage). 

For autologous cell defection to be a viable therapeutic approach, however, it seems preferable to avoid the complication of providing additional growth factor. The drive must be strong enough that the non-producer cells can spread in the population solely because of the growth factor produced by the WT cells. As we have seen, this is determined by the cost/benefit ratio of growth factor production. In addition, the collapse of cooperation must have a significant effect on tumor growth. Successful autologous cell defection, therefore, in which a KO clone spreads to fixation and induces a reduction in overall tumor fitness, requires both a high benefit and a high cost/benefit ratio. In other words, the growth factors must be essential for the tumor, and the cost of producing it must be high enough. Identifying such growth factors and the cancer types (or even the individual patients) in which these conditions apply is a challenge that was not addressed here. 

### 3.3. Differences across Cell Types and Growth Factors

In U266B1 (multiple myeloma) and βTC (neuroendocrine pancreatic cancer), the cost/benefit ratio of the growth factor that was knocked out (respectively, IL6 and IGF2) was high enough to enable the KO clone to spread in the WT population and induce the collapse of cooperation, at least at high FBS. However, while in βTC, the lack of IGF2 was enough to slow down growth significantly, in U266B1, the KO population was still able to proliferate, albeit less efficiently, in the absence of IL6. In NCIH28 (mesothelioma), even though the potential effect of PDGFD deprivation is substantial, its cost/benefit ratio seemed low enough to prevent the invasion of the KO clone. In A549 (lung cancer), instead, while the benefit of IL6 was not high enough to prevent the KO clone from growing, the cost was low enough that the cost/benefit ratio was high enough to drive the spread of the KO clone, but the result was a coexistence of the two clones without a substantial reduction in average cell growth.

Clearly, even when the same growth factor is knocked out, there is no reason to expect that different cell lines will have the same cost/benefit ratio even for the same growth factor (as we indeed observed for IL6 in U266B1 and A549), due to different expression levels. Moreover, each growth factor has different effects on cell fitness, based on the amount of growth factors produced by each cell (which affects the inflection point of the benefit function, *h*), its synergistic effects (which affects the steepness of the benefit function, *s*), its diffusion range (which affects group size, *n*) and how essential it is for cell fitness (which affects the benefit *b*). 

#### 3.3.1. β. TC-ΔIGF2

The effect of IGF2 on βTC cells is well-known. IGF2 is normally expressed only during development [35] but is upregulated in RIP1-Tag2 insulinomas (the source of βTC cells), where it promotes proliferation [36] and protects against apoptosis [37]. IGF2-KO Rip1-Tag2 transgenic mice (the source of our βTC-ΔIGF2 cells) develop fewer, smaller, less aggressive tumors [36,38,39,40,41]). This is in line with the results reported here: a high cost and benefit for IGF2 in βTC cells, and the fact that that βTC-ΔIGF2 can spread among WT cells and lead to a reduction in fitness (a dynamics that was observed previously [20] but without the cost/benefit analysis reported here). A potentially confounding issue is that βTC-ΔIGF2 cells are not autologous to βTC cells but were derived [36] from RIP1-Tag2 IGF2 KO mice [38] in which exon 2 of the IGF2 gene was replaced with the neomycin phosphotransferase gene [42,43]. As βTC and βTC-ΔIGF2 cells are not autologous, the lack of IGF2 production is, arguably, not the only difference between the WT and KO clones. IGF2 KO βTC cells, for example, have smaller nuclei and more cytoplasm. The other cell lines used here, instead, differ from the corresponding WT clones only for the lack of the targeted growth factor gene.

#### 3.3.2. U266B1-ΔIL6

In U266B1, IL6 is known to be overexpressed [44,45], and an autocrine loop seems widely accepted, although paracrine effects are possible [46] and IL6 can in part derive from stromal cells. IL6 is essential in multiple myeloma [47,48,49]: cell lines that do not produce IL6 require exogenously added IL6 [50], antisense IL6 strongly reduces proliferation [51] and blocking the IL6R signaling pathways induces apoptosis while its constitutive activation confers resistance to apoptosis [52]. IL6 promotes cell proliferation in other hematological malignancies [53,54,55] and in solid tumors [56], as it is produced by a variety of cell types and is involved in immune response, inflammation and hematopoiesis [57,58]. This known overexpression of (and dependency on) IL6 in malignant plasma cells is in line with our results that a U266B1-ΔIL6 clone can spread in a WT population; U266B1-ΔIL6 cells, however, can persist and proliferate (albeit slowly) even in the absence of IL6. One notable difference between U266B1 and the other cell lines used here is that U266B1 cells grow in suspension rather than in monolayers. This makes cell growth much more dependent on cell density than for cells growing in monolayers, and growth at different densities might give different results, as density affects group size *n*, which is a strong determinant of the dynamics in collective interactions for the production of growth factors [59]. 

#### 3.3.3. NCIH28-ΔPDGFD

In NCIH28, while PDGFD does not significantly affect the growth of WT cells, knocking down PDGFD or PDGF receptor significantly inhibits cell growth [60,61]. This suggests that PDGFD expression in WT cells is high enough to activate its receptors at maximal levels, and that PDGFD is important for cell proliferation. This does not mean, however, that the cost of production or the benefit is high. Indeed, from our results, it seems that, while there is a substantial dependency on PDGFD, the cost of production is not high enough to drive the KO clone into a WT population. One possible explanation is that the value of the inflection point of the benefit function for PDGFD in NCIH28-ΔPDGFD cells is high enough that there is no stable mixed equilibrium—one of the possible types of dynamics we described. This, however, seems unlikely given that, as observed, endogenous PDGFD expression in WT cells is high enough to activate its receptors at maximal levels. One possible explanation, and a potential confounding issue, is that, even though it codes for a short (370 aa) protein, the PDGFD gene is very long (257k nt), and the NCIH28-ΔPDGFD clone has a very long deletion, which might have deleterious effects that compensate for the advantage of the lack of cost of producing such a long transcript.

#### 3.3.4. A549-ΔIL6

The effect of IL6 in A549 cells is less clear. There is evidence of low expression levels from earlier studies [62] and recent RNA sequencing data [63]. Some earlier evidence suggested that antisense RNA, but not antibodies against IL6, reduce proliferation, which may lead to concluding that IL6 is beneficial for the cells but is mainly intracellular [64]; yet IL6 is clearly secreted in A549 cells [65], and humanized antibodies have been used for therapy [66]. The fact that a mixed equilibrium is found in the long-term experiments reported here suggests that IL6 is at least in part secreted, but its cost/benefit ratio is not high enough to enable an efficient spread. This, combined with the result that the lack of IL6 does not seem to impair the growth of A549-ΔIL6 cells significantly, suggests that this is not an ideal candidate for autologous cell defection.

### 3.4. Outstanding Issues

In summary, of our four combinations, knocking out IGF2 in βTC seems the most effective (strong drive, strong effect on growth), followed by U266B1 (strong drive but not strong effect on growth); knocking out PDGFD in NCIH28 could be effective but the drive is not strong enough and knocking out IL6 in A549 seems to lack both the necessary potency of drive and effect on growth.

The point of these results, however, was not to identify a candidate for therapy but to provide a proof of concept that autologous cell defection can, in principle, lead to the collapse of intra-tumor cooperation based on principles from evolutionary game theory. While the challenge of identifying appropriate cell types and growth factors remains, the feasibility of the approach—using genetically modified cancer cells to disrupt cooperation—seems validated, in principle, by the results, but some conceptual issues remain.

First, the model assumes a well-mixed population, a reasonable assumption for U266B1 (which grows in suspension) but not for the other cell lines (which grown in monolayers), even though serial passaging of adherent cells resembles the periodic disruption of spatial structure implied by the well-mixed model. A model that takes into account the spatial structure of solid tumors, however, would provide a more realistic analysis and is a desirable improvement over the model presented here. Furthermore, while it is reasonable to assume that the benefit function is a sigmoid function of the fraction of producer cells, the exact shape of this function is crucial for the dynamics, and we do not know the values of its parameters *s* (the steepness) and *h* (the inflection point), nor the value of *n* (group size). The benefit function cannot be estimated simply by measuring the effect of a growth factor as a function of its concentration (a routine procedure); what needs to be measured is that effect as a function of the fraction of WT cells in the group. This could be completed by measuring the growth rates of mixed populations with different fractions of WT cells and fitting the data to the equations for the average fitness of a mixed population. Group size itself, however, a major determinant of the dynamics, depends on the diffusion range of the growth factor, a parameter that is difficult to measure (a measurement in vitro would be in any case virtually irrelevant for the dynamics in vivo, as the value of *n* in a tridimensional structure varies widely with the diffusion range, and differently from a planar structure). 

Second, in vitro results are not necessarily expected to hold up in vivo, and the results presented here should be repeated in an animal model. Apart from the inherent higher complexity of an analogous system in vivo, two issues seem relevant for the dynamics: the benefit and the diffusion range of the growth factors. The benefit of the growth factors measured here in vitro is limited to cell proliferation, but there could be additional benefits in vivo, hence a lower cost/benefit ratio, which would confer a higher fitness to producer cells, but could also increase the dependency on the growth factor for the tumor. The diffusion range of the growth factors would also be different: in a 3D system, the number of cells within the same diffusion range is much higher than in a monolayer; however, the diffusion range itself will not be the same in monolayers and solid tumors. The result of combining these opposite effects is not intuitive and requires further theoretical and animal models.

The main, crucial prediction of models of nonlinear public games applied to cancer dynamics, however, is the existence of a mixed equilibrium for cooperation that collapses above a critical cost/benefit function, which seems validated by the results reported here and supports the idea that genetically modified autologous defector cancer cells could drive the collapse of intra-tumor cooperation.

### 3.5. Implications for Therapy

Let us consider, then, the potential translation to therapy. Using engineered cancer cells for cell therapy has the potential benefit of being self-promoting: rather than targeting growth factors directly to reduce their availability for the tumor, for example, using antibodies (to which resistant mutants can evolve [6]), the logic of autologous cell defection is to harness clonal selection—the very force that, for traditional therapies, leads to the evolution of resistance—to drive the spread of non-producer cells into the tumor. As mutant cells that produce additional growth factors cannot invade a population of non-producers, direct resistance to the therapy cannot evolve, and the result would be evolutionarily stable. There are, however, a significant number of unknowns and potential problems.

The first issue is the need to identify the right growth factors. As we have seen here, a high cost/benefit ratio is essential: the cost of growth factor production must be high enough (relative to the benefit of the growth factor) to drive producer cells to extinction. Gene expression and protein levels, as well as the length of the coding sequence, may be correlated to the cost, but ultimately, this must be tested in experiments such as the ones reported here. Knocking out multiple growth factors might seem to be an obvious way to increase the cost. What matters, however, is the cost/benefit ratio, not the cost alone, and knocking out multiple growth factors will not necessarily increase the cost/benefit ratio in an additive way, even though one might argue that the costs of producing multiple growth factors may be additive and the benefits synergistic. Even if we identify suitable growth factors to knock out, we must also show that the lack of those growth factors slows down tumor proliferation, that is, that the growth factor is essential (the benefit is high). While CRISPR-KO library screening data such as the ones provided by the DepMap consortium [63] are useful to identify dependencies in cancer cell lines, they are not relevant for our purpose for two main reasons. First, pooled screening cannot identify dependencies for secreted factors simply because the KO clone can exploit the growth factors provided by the other cells in the pool. Second, it is not enough to make a functional KO (induced by an indel); while this would remove the benefit of the growth factor, it does not remove the cost of production. As the cost of transcription can be as high as the cost of protein production, to eliminate the cost of growth factor production in the KO cells (the very feature that drives their spread in the population of WT cells), one must prevent transcription and translation, by removing the entire coding sequence (for short genes) or the transcription start site [67,68]. 

The second type of problem is practical access and delivery. First, tumor heterogeneity raises the problem of obtaining a relevant clone to knockout. Second, the primary tumor must be accessible enough for harvesting and re-injecting cells but not enough to be out of reach for removal surgery. Metastases are also a problem: while in principle metastases could be reached by modified cells injected in the bloodstream, this currently seems a remote possibility. Perhaps the method might be more feasible for hematological malignancies than for solid metastatic tumors.

The third type of problem is about the efficacy of the gene drive (will the KO cells spread fast enough?) and of the effect of the collapse of cooperation (will the tumor still grow without the growth factor?). The logic is that the KO clone will spread within the tumor by clonal selection, such as resistant clones spread in the presence of selection induced by therapy. While in our populations in vitro extinction of the KO clone occurred in a relatively short time, whether this is also the case in vivo remains to be seen. Furthermore, in vivo growth factors are also provided by stromal cells in the tumor microenvironment. Stromal cells are activated by growth factors produced by the cancer cells, which must also be knocked out to prevent the continued secretion of growth factors by the stroma. Additionally, even if a KO clone can drive the WT clone to extinction quickly enough to have clinical relevance, it is possible that removing one or even a few growth factors is not enough to stop tumor growth (on the other hand, the lack of growth factors might at least reduce the deleterious effects of cachexia, which is due to the overproduction of secreted factor [69]).

Finally, even though the therapy would be, by design, immune from direct evolution of resistance (because it is driven by clonal selection: WT cells that produce new growth factors cannot invade a population of KO cells), alternative ways for resistance to evolve might exist, for example, the constitutive activation of its downstream pathway would make a growth factor irrelevant. Knocking out multiple growth factors, so at least some pathways will be silenced, would be necessary. 

## 4. Conclusions

The main postulate of autologous cell defection [23,24,25], that a small fraction of KO non-producer cells can drive a population of WT producer cells to extinction, seems validated in some cases. The cases in which KO cells do not lead to the collapse of cooperation, or in which the collapse of cooperation does not lead to a significant reduction in proliferation, can be understood based on the costs and benefits of growth factor production, in line with the predictions of evolutionary game theory. Identifying the appropriate growth factors to knock out in each tumor type or in each individual patient remains a challenge, and many issues about engineering, delivery and efficacy remain to be tested to apply these results to therapy. 

## 5. Materials and Methods

### 5.1. Cell Lines

Three human cancer cell lines were obtained from the American Type Culture Collection: U266B1 (multiple myeloma), A549 (non-small cell lung cancer [70]), NCIH28 (mesothelioma [71]). A fourth cell line, murine βTC (neuroendocrine pancreatic cancer), was a gift from Gerhard Christofori (University of Basel) [20,36]. These cell lines were labeled using a lentiviral vector with ubiquitous PGK promoter driving expression of EGFP (Addgene #21316 PL-SIN-PGK-EGFP).

### 5.2. Gene Knock-Out

Knock-out (KO) clones of the three human cell lines were produced by deleting part of the coding sequence of the gene for a growth factor by delivery of Cas9/sgRNA ribonucleoprotein complexes [72], using pairs of sgRNA targeting two sides of the gene, via electroporation using a 4D Nucleofector X unit (Lonza). sgRNAs were designed to avoid off-target effects (no other site in the genome contains up to 2 mismatches), and deletion was confirmed by sequencing. For U266B1 and A549, the entire coding sequence of the IL6 gene was deleted using the following sgRNAs: GCATGGCAAGACACAACTAG (5′ end); GATCATTTCTTGGAAAGTGT (3′ end), producing clones U266B1-ΔIL6 and A549-ΔIL6, respectively. For NCIH28, part of the PDGFD gene was deleted, including the first exon, using the following sgRNAs: GACCAACTTACTGGAAGTTA (5′ end); GTCCCGGCCGGCGATTAAAC (3′ end) to produce clone NCIH28-ΔPDGFD. A βTC KO clone for the IGF2 gene (βTC-ΔIGF2) was provided by Gerhard Christofori [20,36]. 

### 5.3. Cell Culture

All cell lines were tested (and confirmed negative) for *Mycoplasma* and maintained in culture at 37 °C and 5% CO_2_ in growth medium (for U266B1 and NCIH28: RPMI; for A549: F12K; for βTC: DMEM) with variable amounts of FBS (Fetal Bovine Serum), 1% glutamine and 1% antibiotics (penicillin-streptomycin). Recombinant growth factors, when used, were human IL6 (R&D #206IL), human PDGFD (R&D #1159-SB-025) and murine IGF2 (R&D #792MG). For short-term culture experiments to measure relative growth rates, cells were grown with 5% FBS in 6-well plates (surface 9 cm^2^) starting from 2 × 10^5^ cells until one of the wells became confluent (or, for U266B1, when density reached 4 × 10^6^ cells/mL); at that point, all wells were measured (growth rates are reported as cell number relative to the well with the fastest growth). For long-term culture experiments, cells were grown in flasks (surface 25 cm^2^) and passaged to new flasks when confluent or, for U266B1, split (50% growth medium replaced with fresh medium) when density reached 4 × 10^6^ cells/mL. The fraction of the two types in mixed populations was measured by flow cytometry using an Accuri C6 flow cytometer (Becton Dickinson) (Ex 488 nm; Em 533/30 nm BP, emission filter FL1) after gating out cellular debris and selecting only single cells for analysis (50,000 cells per count). Differences in growth rates were analyzed using Student’s t-test or Welch’s unequal variances t-test, depending on the sample variance. 

### 5.4. Mathematical Model

Interactions between cells are modeled using a public goods game in the framework of evolutionary game theory [26,73]. In a group of cells, a public good (a benefit equal to 1) is produced by the growth factors shared by the WT cells. The size of the group (*n*) is defined by the diffusion range of the growth factors, which is supposed to be secreted and diffusible in the extracellular matrix. There are two types of cells: producers (WT, cooperators) and non-producers (KO, defectors). The benefit accrues to both types. Only the WT cells, however, pay the cost (*c*) of producing the growth factor. In a large population with random group formation at each generation, the probability of having *i* WT cells among the other cells in the group, given a frequency *x* of WT cells in the population, is given by the probability mass function of a binomially distributed random variable *i* with parameters (*n* − 1, *x*):$$p_{i,n-1}\left(x\right)=\left(\begin{array}{c} n-1\\ i \end{array}\right)x^{i}{\left(1-x\right)}^{n-1-i}$$

The fitnesses of *WT* and *KO* cells are given, therefore, by
$$W_{WT}=W_{0}+\overset{n-1}{\underset{i=0}{\sum }}p_{i,n-1}\left(x\right)\cdot b\left(i+1\right)-c$$
$$W_{KO}=W_{0}+\overset{n-1}{\underset{i=0}{\sum }}p_{i,n-1}\left(x\right)\cdot b\left(i\right)$$
where *b*(*i*) is the benefit for being in a group with *i* other *WT* cells and *W*_0_ is the baseline fitness. The average fitness of a mixed population is
$$\bar{W}=x\cdot W_{WT}+\left(1-x\right)\cdot W_{KO}$$

Clonal selection can be described by the replicator dynamics [74] of this system:$$\dot{x}=x\;\left(1-x\right)\;\beta \left(x\right)$$
where β(*x*) is the fitness difference *W_WT_*-*W_KO_*:$$\beta (x)=\sum_{i=0}^{n-1}p_{i,n-1}\left(x\right)\cdot b\left(i+1\right)-c-b\left(i\right)$$

Assuming, as is generally the case, that the effect of growth factors is a nonlinear function of its concentration (hence of the faction of WT cells), the benefit *b*(*i*) for an individual in a group with *i* WT cells and (*n* − *i*) KO cells is given by
$$b\left(i\right)=\frac{\left[l\left(i\right)-l\left(0\right)\right]}{\left[l\left(n\right)-l\left(0\right)\right]}$$
the normalized version of a logistic function *l*(*i*) with inflection *h* and steepness *s*:$$l\left(i\right)=\frac{1}{1+e^{s\left(h-i/n\right)}}$$

The parameter *h* defines the position of the inflection point: *h*→1 gives increasing returns (convex) and *h*→0 diminishing returns (concave); *s* defines the steepness of the function (*s*→∞ models a threshold public goods game; *s*→0 models a linear public goods game; the normalization prevents the logistic function from becoming constant for *s*→0). The model, therefore, can be used to model growth factors for which the effect on cell fitness is a concave, convex, threshold or sigmoid function of its concentration.

## Figures and Tables

**Figure 1 cancers-13-03674-f001:**
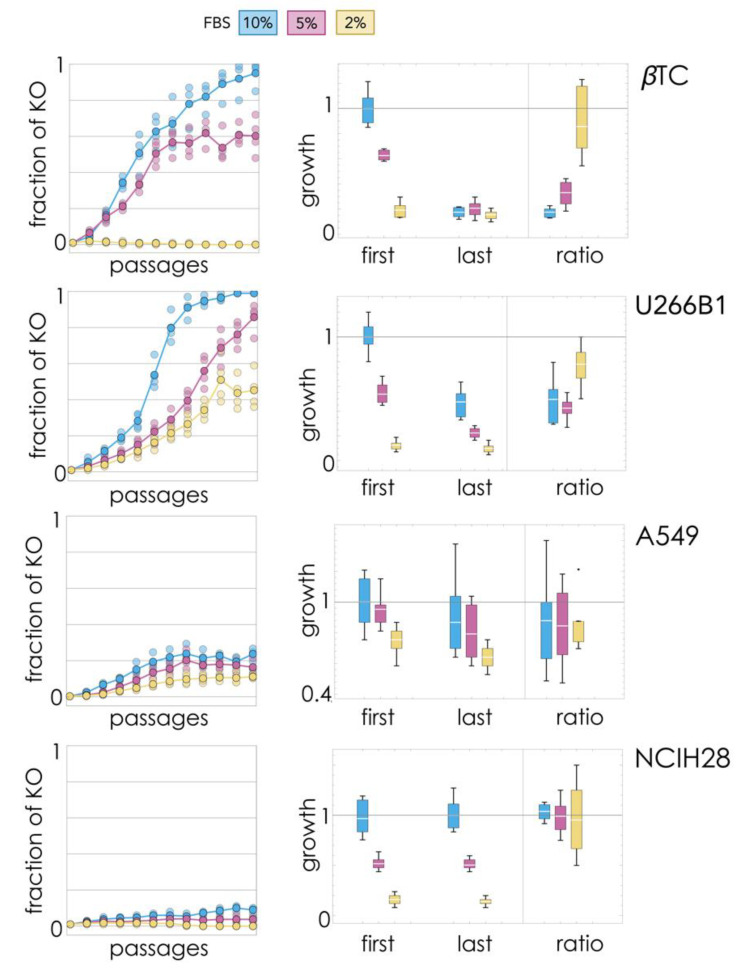
Change over time in the fraction of KO cells and relative growth of populations when 1% KO cells are introduced in a population of the WT cell line, with different amounts of FBS. The left panels show the fraction of KO cells in each population (light dots) and the mean value of four populations (darker dots connected by a line) at each passage. The right panels show the growth rates (relative to the average maximum growth rate) and their ratio for the first and last passage.

**Figure 2 cancers-13-03674-f002:**
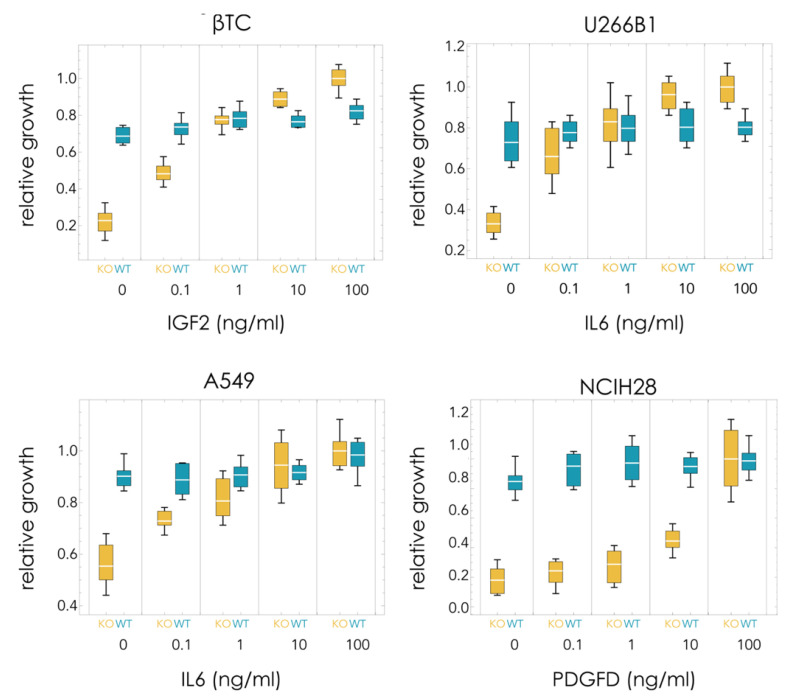
Growth rates of pure WT or KO populations with different amounts of exogenous growth factor (5% FBS).

**Figure 3 cancers-13-03674-f003:**
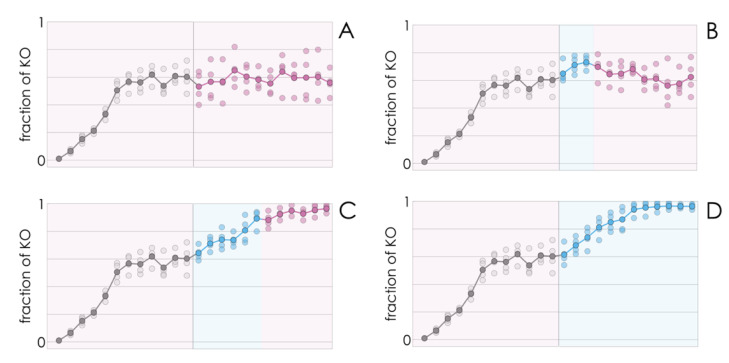
A transient increase in the amount of FBS can lead to stable collapse of cooperation. βTC cells (WT) coexist with βTC-ΔIGF2 (KO) cells after about 6 passages with 5% FBS (gray dots—see Figure 1). (**A**) The population remains mixed for 18 passages. (**B**) If FBS is increased to 10% for only three passages, the KO clone slightly increases in frequency, but the population then returns to the original mixed equilibrium (**C**) If FBS is increased to 10% for six passages, the KO clone increases in frequency even after FBS is reduced to 5% and drives the WT clone to extinction. (**D**) If FBS is increased to 10%, the KO clone (blue dots) increases in frequency and drives the WT clone to extinction.

**Figure 4 cancers-13-03674-f004:**
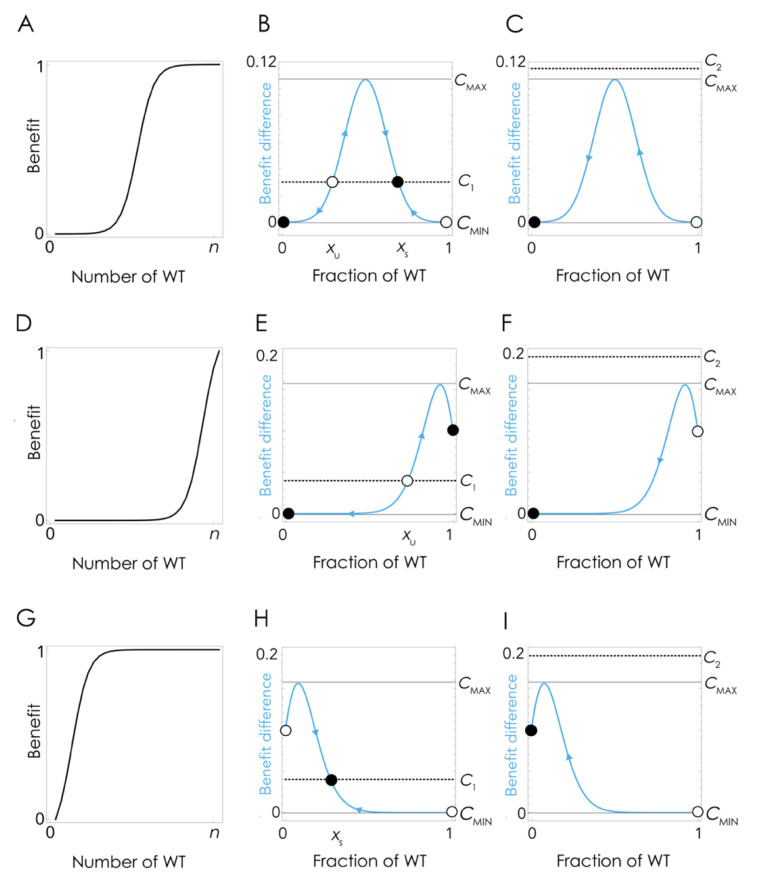
Evolutionary dynamics of cooperation for the production of growth factors. The black curves show the effect (benefit) of the growth factor as a function of the number of WT cells in the group (of size *n*). The blue curves show the difference in benefit between WT and KO cells: W_WT_-W_KO_-c (assuming *b* = 1); when this is greater than the cost c, the fraction of WT cells increases in frequency. Hence, arrows show the direction of the dynamics, and circles show the equilibria (black: stable; white: unstable); The two mixed equilibria are labeled x_S_ (stable) and x_U_ (unstable). (**A**) Since growth factors have a nonlinear effect on cell proliferation, we assume that the benefit is a sigmoid function of the fraction of WT cells; here, *n* = 30, *s* = 30, *h* = 0.5. (**B**) If the cost/benefit ratio of the growth factor is below a critical level c_MAX_, for example, c_1_, a WT population can be invaded by a KO clone and converge to a stable polymorphic equilibrium. (**C**) If the cost/benefit ratio is higher than c_MAX_, for example, c_2_, cooperation is not stable, and the KO mutant will increase in frequency until the WT clone is extinct. If the cost/benefit ratio returns to c_1_ when the fraction of WT cells is below the unstable equilibrium x_U,_ it will continue to decline to the pure equilibrium 0 even though the original mixed equilibrium is stable again. (**D**–**F**) Same as (**A**–**C**) but with a higher inflection point (*h* = 0.9). (**G**–**I**) Same as (**A**–**C**) but with a lower inflection point (*h* = 0.1).

## Data Availability

The data presented in this study are available in the Appendix A.

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
