# Peer review of "Collapse of Intra-Tumor Cooperation Induced by Engineered Defector Cells"

_cancers, 2021, doi:10.3390/cancers13153674_

Round 1
Reviewer 1 Report
Review in attachment.

Reviewer 2 Report
Very well presented experimental exploration of tumor growth dynamics and therapy resistance based on game theory.
Minor spell checks required (e.g. in Abstract: "An effective therapy must be immune from invasion by such mutant cells."; the meaning is not clear enough and this is a crucial sentence for the introduction of readers into the concept).
Reviewer 3 Report
There are extensive published literatures reporting on how drug resistance may arise due to the phenomenon of clonal selection, whereby drug treatment confers higher fitness to rare mutant cancer cells that consequently proliferate to become the dominant subclones. However, evidence of widespread intra-tumour heterogeneity observed after drug treatment (e.g. Almendro, V. et al. Cell Reports 6, 514–527 (2014); Caswell-Jin, J. L. et al. Nat Commun 10, 657 (2019); Stewart, C. A. et al. Nat Cancer 1, 423–436 (2020).) suggest that subclones with lower fitness persist within tumours, and raises the possibility that intra-tumour cooperation between subclones with different fitnesses drives the development of drug refractoriness. Little is hitherto known about the nature and dynamics of such intra-tumour cooperation between cancer cells. Herein, Archetti's current manuscript studies the evolutionary dynamics of cooperating cancer cells using evolutionary game theory, and describes how perturbation to such dynamics may constitute a plausible therapeutic option that is potentially less drastic than the conventional "slash-and-burn" maximum dosage chemotherapeutic treatment regime and is likely to avert strong selection of resistant mutant cells. Archetti's approach is highly novel in the following ways:
(i) Concept: Unlike the adaptive therapy championed by Robert Gatenby that focuses on cancer subclones in direct competition (resistant versus non-resistant subclones) with each other, Archetti studies two subclones of cancer cells (producer versus non-producer cells) that not only compete against each other but also cooperate with each other via sharing of secreted public goods. The under-studied phenomenon of intra-tumour cooperation is thus factored into Archetti's study when considering how manoeuvring the growth condition may alter the ratio of the two subclones in achieving a retarded tumour growth as the ideal therapeutic outcome;
(ii) Methodology: In considering intra-tumour cooperation between cancer cells, Archetti mobilises the use of evolutionary game theory in modelling the evolutionary dynamics of the producer/non-producer cells based on the cost-benefit calculations derived from empirical observations. This differs significantly from conventional cancer therapy papers that use an arsenal of molecular and/or genetic techniques to reach their conclusions and generally shy away from the use of theoretical tools to model complex social behaviours of the heterogenous cancer cells that may also contribute to drug refractoriness of the tumour-at-large.
(iii) Cell Lines: The use of 4 different cell lines with different permutation of cost/benefit attributes has been highly useful in illuminating the different trajectories possible in modelling potential therapeutic effect of growth factor addition and/or KO cell addition, and provides a highly useful framework in considering the appropriate KO target with respect to the cancer type under study for Archetti’s therapeutic proposal.
The author should take note of the following minor points in revising the manuscript:
(a) The reasons behind the selection of KO targets for each cell line tested (IGF2 for BTC cell line, IL6 for U26B1/A549, PDGFD for NCIH28) should at least be briefly mentioned under "introduction" or "result" section with more details being provided under the M&M section, so that readers wont be left wondering the justification for the KO target for each cell line, especially for readers who are not acquainted with the specific growth factor and/or cancer cell line used.
(b) The lack of an animal model (e.g. mice xenograft) as proof-of-concept of the author’s therapeutic ideas, though I personally feel that the manuscript is already full of data churned out using cell line models that readers will need to grapple with. The author can at least address/predict how the outcome may be like with the use of an animal model, when compared to cell lines (adherent / suspension). It is well known that chemotherapeutic drugs may not be able to penetrate the tumour, especially in areas not well-served by blood vessels. How can the exogenous application of growth factors or delivery of KO cancer cells be affected in view of issue relating to difficulties in tumour penetration? I must add that the author actually gave a very good description regarding issues of tumour access and delivery (line 503 to 509)
(c) The manuscript will benefit tremendously if the author can sub-divide the “Discussion” into sections with sub-headings that appropriately describe the various sections. The author should also consider summarising specific parts of the “Discussion” section to avoid fatigue in trying to comprehend the complex logic behind the different trends observed and possible explanations. Specifically, the paragraphs describing each growth factor in details in relation to the cancer type/cell lines used (line 368 – 427) is lengthy and should be curtailed substantially.
Other very minor mistakes:
- Line 244 – “The coexistence of the two clones at of lower amounts of FBS can be explained by the……” to remove the word “of”
- Line 330-331 – “. In our case, this is due to the reduction of the frequency of KO cells induced by the increase in available growth factor…” Is this supposed to be increase rather than reduction of KO cells?
